# microRNA-146a-5p, Neurotropic Viral Infection and Prion Disease (PrD)

**DOI:** 10.3390/ijms22179198

**Published:** 2021-08-25

**Authors:** Aileen I. Pogue, Walter J. Lukiw

**Affiliations:** 1Alchem Biotech, Toronto, ON M5S 1A8, Canada; wlukiw@yahoo.com; 2LSU Neuroscience Center, Louisiana State University Health Science Center, New Orleans, LA 70112, USA; 3Department of Ophthalmology, Louisiana State University Health Science Center, New Orleans, LA 70112, USA; 4Department of Neurology, Louisiana State University Health Science Center, New Orleans, LA 70112, USA

**Keywords:** aging, Alzheimer’s disease, bovine spongiform encephalopathy (BSE), Creutzfeldt–Jakob disease (CJD), Gerstmann–Sträussler–Scheinker syndrome (GSS), microRNA-146a, NF-kB (p50/p65), prion disease, reactive oxygen species (ROS), scrapie, SARS-CoV-2

## Abstract

The human brain and central nervous system (CNS) harbor a select sub-group of potentially pathogenic microRNAs (miRNAs), including a well-characterized NF-kB-sensitive *Homo sapiens* microRNA hsa-miRNA-146a-5p (miRNA-146a). miRNA-146a is significantly over-expressed in progressive and often lethal viral- and prion-mediated and related neurological syndromes associated with progressive inflammatory neurodegeneration. These include ~18 different viral-induced encephalopathies for which data are available, at least ~10 known prion diseases (PrD) of animals and humans, Alzheimer’s disease (AD) and other sporadic and progressive age-related neurological disorders. Despite the apparent lack of nucleic acids in prions, both DNA- and RNA-containing viruses along with prions significantly induce miRNA-146a in the infected host, but whether this represents part of the host’s adaptive immunity, innate-immune response or a mechanism to enable the invading prion or virus a successful infection is not well understood. Current findings suggest an early and highly interactive role for miRNA-146a: (**i**) as a major small noncoding RNA (sncRNA) regulator of innate-immune responses and inflammatory signaling in cells of the human brain and CNS; (**ii**) as a critical component of the complement system and immune-related neurological dysfunction; (**iii**) as an inducible sncRNA of the brain and CNS that lies at a critical intersection of several important neurobiological adaptive immune response processes with highly interactive associations involving complement factor H (CFH), Toll-like receptor pathways, the innate-immunity, cytokine production, apoptosis and neural cell decline; and (**iv**) as a potential biomarker for viral infection, TSE and AD and other neurological diseases in both animals and humans. In this report, we review the recent data supporting the idea that miRNA-146a may represent a novel and unique sncRNA-based biomarker for inflammatory neurodegeneration in multiple species. This paper further reviews the current state of knowledge regarding the nature and mechanism of miRNA-146a in viral and prion infection of the human brain and CNS with reference to AD wherever possible.

## 1. Introduction and Overview

Multiple independent research laboratories have reported the significant upregulation of a specific sub-group of pathogenic microRNAs (miRNAs) in progressive, neuro-inflammatory, incapacitating and often lethal neurodegenerative diseases of the human brain and CNS. As soluble, amphipathic regulatory molecules, miRNAs are important post-transcriptional and epigenetic regulators of messenger RNA (mRNA) abundance, speciation and complexity [1,2]. These microRNAs: (**i**) exist as ~18- to ~25-ribonucleotide (nt), single-stranded noncoding RNAs (sncRNAs) whose sequences are both unique and highly selected over evolution; (**ii**) represent the smallest information-carrying ribonucleic acids yet defined; (**iii**) have been repeatedly shown to play critical and determinant roles in the onset and propagation of many human CNS disorders, including progressive, incapacitating and terminal neurological syndromes; and (**iv**) are fascinating molecular entities because of their somewhat unconventional origin, their ribonucleotide sequence characteristics, their ability to regulate multiple cellular processes in health and disease, their immense potential in disease therapeutics and evolutionary dynamics [1,3,4,5,6,7]. Regarding the evolutionary aspects of miRNA and miRNA sequence complexity and selection, from mathematical considerations alone, a ~22 nt sncRNA with the possibility of four ribonucleotides at each position (A, C, G and U—adenine, cytosine, guanine and uracil, respectively) has the potential to generate *an exponentiation of 4 to the power of 22* or about 1.76 × 10^13^ unique miRNA species. However, genome-wide analysis of all known human miRNAs, with a current total of ~2650 individual species, indicates that the RNA sequence of human miRNAs has been highly selected from many different RNA sequence possibilities. Interestingly, some miRNAs have been shown to maintain their exact or highly homologous miRNA ribonucleotide sequence between plants and animals over ~1.5 billion years of evolution (the *Arabidopsis thaliana*–*Homo sapiens* divergence), indicating that highly conserved ribonucleotide sequence-mediated genetic regulatory functions are attributable to the same miRNAs and miRNA binding proteins over vast periods of time [1,8,9]. Further RNA-sequencing and array-based analyses have indicated that only certain miRNAs, probably about 25–30 individual miRNA species: (**i**) are abundant in the cytoplasm of the human brain as well as retinal and other CNS cells; (**ii**) are inducible by pathological factors, such as pro-inflammatory cytokines and chemokines; and (**iii**) are upregulated by different types of pathogenic microbes, including viral gene-encoded products and highly neurotoxic secreted bacterial exudates such as lipopolysaccharide (LPS) [5,10,11] see below.

## 2. *Homo sapiens* miRNA-146a-5p (hsa-miRNA-146a-5p) and Mechanism of Action

Amongst these brain- and CNS-expressed microRNAs, one of the most well-studied microRNA species known to be involved in the pathogenesis of progressive, age-related neurological diseases is an inducible *Homo sapiens* microRNA-146a-5p (hsa-miRNA-146a-5p; hsa-miRNA-146a; or simply miRNA-146a) whose significant upregulation is currently implicated in every prion disease (PrD) in humans and animals analyzed to date [12,13,14] (see below); in at least 18 viral-induced encephalopathies, Alzheimer’s disease (AD) [15,16,17] and multiple neurological disorders that include cerebrovascular disease (CVD) [3,18], traumatic brain injury (TBI) [19], temporal lobe epilepsy (TLE) [20], age-related macular degeneration (AMD) [10,11,21,22], amyotrophic lateral sclerosis (ALS) [3,23], peripheral neuropathies and neurological tumors of the CNS [3] and neuro-immune diseases such as myasthenia gravis (MG) and multiple sclerosis (MS) [24] (Table 1).

In both *Homo sapiens* and *Mus musculus* microRNA-146a-5p (hsa-miRNA-146a and mmu-miRNA-146a, respectively) are 22-nucleotide (40.9%G+C) single-stranded small noncoding RNAs (sncRNA; 5′-UGAGAACUGAAUUCCAUGGGUU-3′; miRBase Acc-ession MIMAT0000449 and MI0000170, respectively; https://www.mirbase.org/cgi-bin/mature.pl?mature_acc=MIMAT0000449; http://www.mirbase.org/cgi-bin/mirnaentry.pl?acc=MI0000170; https://www.Genecards.org/cgi-bin/carddisp.pl?gene=MIR-146A last accessed on 23 August 2021; ad-ditional hsa miRNA-146a-5p sequence and associated disease-relevant biological data in AD are also available at Alzheimer Disease disease: Malacards—Research Articles, Drugs, Genes, Clinical Trials; https://www.malacards.org/card/alzheimer_disease?search=146a; last accessed on 23 August 2021).

Human and murine miRNA-146a are amongst the most intensively studied sncRNAs in all human and murine neurobiology, neuropathology, neurodegenerative disease and transgenic (Tg) murine models of these diseases. The equivalent and preserved 100% identical RNA sequence of miRNA-146a between *Homo sapiens* and *Mus musculus*: (**i**) attests to the basal importance of this sncRNA from an evolutionary perspective and (**ii**) is highly useful in molecular-genetic comparative and modelling studies and considerations, because this sncRNA has an identical RNA sequence in both species. Encoded from a single locus at chromosome 5q33.3 in humans (https://www.genecards.org/cgi-bin/carddisp.pl?gene=MIR146A accessed on 16 August 2021; human genome version GRCh38/hg38) and at chromosome 11q in mice, human miRNA-146a-5p has three tandem NF-KB (p50/p65) binding sites in its immediate upstream 5′ promoter. This miRNA can be rapidly induced by NF-kB (50/p65) and pathological mechanisms that upregulate NF-kB, such as biophysical and biochemical inducers, including hypoxia, reactive oxygen species (ROS), bacterial endotoxins and lipopolysaccharides (LPS), amyloid peptides, cytokines and other cell stressors [63,64,65]. Importantly miRNA-146a has a relatively short half-life of about 1.5–2 h in human brain cells and tissues; however, the stability of this sncRNA may be extended under nonphysiological circumstances and/or in certain subcellular compartments [39,65].

miRNA-146a, a unique, relatively abundant and distinctive member of the microRNA gene family, was originally described as being significantly induced and upregulated after microbial endotoxin, lipopolysaccharide (LPS) and/or cytokine stimulation of human THP1 cells, monocytes originally derived and immortalized from an acute monocytic leukemia patient of the M5 subtype [7,66]. THP1 cells are microglia-like and phagocytic for both latex beads and sensitized erythrocytes, lack cell surface and cytoplasmic immunoglobulins and are involved in the clearance of the 42 amino acid amyloid beta (Aβ42) peptide, a major biomarker for AD. It was subsequently found that the induction of this endotoxin-responsive microRNA is under transcriptional control by NF-kB (p50/p65); shortly thereafter, this inducible, pro-inflammatory miRNA-146a was found to be upregulated by metal sulfate-generated reactive oxygen species (ROS); by pro-inflammatory cytokines, such as IL-1β and TNFα; by bacterial endotoxins, such as LPS and fragilysin; by Aβ42 peptides; by inflammatory cocktails containing IL-1β and Aβ42 peptides together, in stressed human primary neuronal–glial (HNG) cells in primary culture, in stressed human brain-derived microglial (HMG) cells; and by many different strains of prions and neurotropic viruses [7,16,21,40,53,63,66] (see Table 1). Importantly, miRNA-146a is also found to be moderately abundant in the aging human brain and CNS and the immune cells of mice and humans where its over-expression during neurodegenerative disease contributes to astroglial proliferation and astrogliosis, cytokine overexpression, deficits in the innate-immune response and the initiation of inflammatory events leading to dysfunctional neurons, synaptic deficits and eventually neuronal cell atrophy and brain cell death [13,21,40,55,67].

Similar to all microRNAs, the major mechanism of miRNA genetic and neurobiological activity is to ‘seek out’ and interact via base pair recognition, complementarity and noncovalent hydrogen binding within the 3′-untranslated region (3′-UTR) of its target mRNA 3′-UTRs, and, in doing so, it decrease the capability of the specific mRNA to be expressed [2,68,69]. All metazoan miRNAs appear to initially recognize their target mRNAs by recognition of a ‘seed region’ or ‘seed sequence’ in the 3′-UTR; this ‘seed region’ is a conserved heptametrical sequence co-localized at positions 2–7 from the 5′-end of the miRNA. Even though base pairing of miRNA and its target mRNA often does not perfectly match, the ‘seed sequence’ is perfectly complementary [2,5,11]. In human neurodegenerative disease, miRNA-146a has several known relevant and verified targets that include the 3′-UTRs of: (**i**) the complement factor H (CFH) mRNA involved in inflammation and the innate immune response; (**ii**) the membrane-spanning TSPAN-12 protein involved in amyloidogenesis and the clearance of amyloid beta peptides from brain cells; and (**iii**) the interleukin-1 receptor kinase IRAK-1 (with a compensatory increase in IRAK-2) involved in pathological NF-kB accumulation, signaling and neuro-inflammation [2,68,69,70,71]. Upregulated miRNA-146a has been definitively linked to the downregulation in the expression of CFH, TSPAN-12 and IRAK-1 both in HNG cells in primary culture and in AD brain [10,11,64,69].

## 3. Neurotropic Viral-mediated Induction of hsa-miRNA-146a-5p

One truly remarkable and widely observed phenomenon concerning hsa-miRNA-146a-5p is that this inducible sncRNA is significantly upregulated by at least 18 neurotropic DNA and RNA viruses that infect the human brain, CNS, immune, lymphatic and hepatic and/or circulatory systems. In alphabetical order, these include (**i**) Borna encephalitis disease virus 1 (BoEDV-1; BDV; *Mononegavirales;* (−)ssRNA genome; [26,27]; (**ii**) Chikungunya virus (CHIKV; *Togaviridae;* (+)ssRNA genome; [28]; (**iii**) enterovirus 71 (EV71; *Picornaviridae*; (+)ssRNA genome; [29]; (**iv**) Epstein–Barr virus (EBV; *Herpesviridae*; dsDNA genome; [30]; (**v**) hantavirus (HTV; *Bunyaviridae;* (−)ssRNA genome; [31]; (**vi**) hepatitis A virus (HAV; *Picornaviridae;* (+)ssRNA genome; [32]; (**vii**) hepatitis B virus (HBV; *Hepadnaviridae*; dsDNA genome; [33,34,35,36,72]; (**viii**) hepatitis C virus (HCV; *Flaviviridae*; (+)ssRNA genome; [37]; (**ix**) herpes simplex virus-1 (HSV-1; *Herpesviridae*; dsDNA genome; [38,39,41]; (**x**) Hendra (*Henipavirus*) virus (HeV; *Paramyxoviridae*; (−)ssRNA genome; [42]; (**xi**) human immunodeficiency virus (HIV; *Retroviridae*; (+)ssRNA genome; [43]; (**xii**) human influenza A viruses (H1N1/H3N2; *Orthomyxoviridae*; (+)ssRNA genome; [44,45]; (**xiii**) early human papillomavirus virus 16 (eHPV-16; *Papillomaviridae;* dsDNA genome; [46]; (**xiv**) human severe acute respiratory syndrome coronavirus-2 (SARS-CoV-2; the causative agent of COVID-19 disease; genus *Betacoronavirus* of the family *Coronaviridae; (+)ssRNA genome;* [47]; unpublished data; (**xv**) human T-cell leukemia (lymphotropic) virus type 1 (HTLV-1; *Retroviridae*; (+)ssRNA genome; [48]; (**xvi**) Japanese encephalitis virus (JEV; *Flaviviridae;* (+)ssRNA genome; [49,50]; (**xvii**) Kaposi’s sarcoma-associated herpesvirus (KSHV; *Herpesviridae*; dsDNA genome; [51]; and (**xviii**) severe fever with thrombocytopenia syndrome virus (SFTSV; *Bunyaviridae;* (−)ssRNA genome; [27,52] (Table 1). Whether the significant induction of host miRNA-146a after viral invasion and/or successful infection is a protective mechanism of the human host cell or a strategy used by the virus for invasion and productive replication is currently not well understood [16,21,35,40,49,53,73,74]. Regardless of its innate-immune, inflammatory signaling, pro- or anti-viral signaling function miRNA-146a has been shown to be in part compartmentalized and packaged into exosomes (EXs) and/or extracellular microvesicles (EMVs); after these EXs and/or EMVs are released from donor cells, they may be taken up by recipient cells and function in the modulation and/or spread of miRNA-146a-mediated gene expression signaling during either viral or prion infection [35,53]. Interestingly, all of the many types of viral infections that induce miRNA-146a are associated with specific neurological disease symptoms and/or syndromes that are progressive, age-related, insidious, incapacitating and often lethal. It is clear that the ubiquity of miRNA-146a upregulation in neurotropic viral infection indicates that this pro-inflammatory miRNA lies at a critical intersection of several important neuro-biological immune-response processes with highly interactive associations affecting the following: (**i**) Toll-like receptor signaling pathways; (**ii**) the inflammatory and innate- immune response; (**iii**) cytokine storms involving cytokine IL-1β, TNFα and chemokine production, most often as complex pro-inflammatory cocktails in neural cells and tissues; (**iv**) morphological change in neuronal cytoarchitecture and apoptosis; and (**v**) neural cell decline and neuronal cell demise [4,16,45,53,75,76].

Recent evidence has indicated that, in general, endogenous human miRNAs (small, single-stranded ~22 nt RNAs, such as miRNA-146a) might have a significant antiviral role and hence may be useful to the host in an innate-immune defense by targeting the single-stranded viral ribonucleic acid (ssvRNA) genomes of several different neurotropic viruses, such as SARS-CoV-2, and function to downregulate or modulate their expression [77,78]. To this end, and to quote just one recent example, RNA sequencing and other analytical genetics in RNA-based sequence studies have shown that at least ~160 of the ~2650 of the 18–22 nt naturally occurring human single-stranded miRNAs have perfect complementarity within the miRNA-mRNA ‘*seed region*’ of the SARS-CoV-2 ssvRNA genome, and these include the unique hsa-miRNA-146a ribonucleotide sequence and those of other related microRNA gene families [47,73,79,80,81]. Interestingly, the apparent lack of nucleic acids detectable within the prion particle indicates that some yet poorly understood pathological mechanism is responsible for the prion-mediated upregulation of miRNA-146a when neurons are confronted with infection by these neurological-disease inducing sialoglycoproteins [16,53,60,75,82,83].

## 4. Prion Disease (PrD) Upregulates hsa-miRNA-146a-5p

Prion diseases (PrD), coined from the term ‘proteinaceous infectious particle’ represent a group of progressive, transmissible and incurable spongiform encephalopathies (TSEs) comprising a small family of relatively rare, rapid onset and consistently fatal neurodegenerative disorders affecting both animals and humans [25,75]. TSEs are characterized by: (**i**) decreased activity of cholinergic and gamma-amino butyric acid (GABA) pathway-related enzymes, while adrenergic pathways are relatively spared; (**ii**) distinctive spongiform changes in the neocortex and other anatomical areas of brain tissues; (**iii**) association with long prodromal and incubation periods; (**iv**) synaptic and dendritic damage and dysfunctional connectivity in the neocortex; (**v**) progressive neuronal atrophy and loss; (**vi**) gliosis and the unusually rapid proliferation of neuroglial cells; (**vii**) an atypical inflammatory response in neural tissues, which further stimulates spongiform change; and (**viii**) the induction of sncRNAs, including miRNA-146a [3,16,53,59,60,75,83,84,85]. Naturally occurring TSEs of herbivorous mammals of the orders *Ruminantia* and *Artiodactyla* have been considered the ‘prototype’ of prion disease, and these include scrapie of sheep and goats (both of the family *Bovidae*, subfamily *Caprinae*) and bovine spongiform encephalopathy (BSE, or ‘mad cow disease’) of cattle (family *Bovidae*, subfamily *Bovinae*). These animal diseases of the family *Bovidae* are closely related in their molecular and genetic neurobiology, pathology and clinical presentation to the human neurological disorders Gerstmann–Sträussler–Scheinker syndrome (GSS) and Creutzfeldt–Jakob disease (CJD) [3,16,25,53,56,57,58,59,75,86,87]; https://www.nhs.uk/conditions/creutzfeldt-jakob-disease-cjd/; last accessed on 23 August 2021; see Table 1).

Largely because of their unusual and novel nature, PrDs have been intensively studied and are known to be caused by a misfolded isoform of a ubiquitous and highly conserved brain-, CNS- and PNS-enriched cellular prion sialoglycoprotein known as PrPc. The PrPc monomer is a ~209 amino acid (~200 kDa) glycosylated cell surface polypeptide, containing a predominant internal α-helical region, encoded in humans at chr 20p13. Normally, the constitutively expressed PrPc appears to be involved in neuritogenesis, neuronal homeostasis, cell signaling, cell–cell adhesion and interaction and intercellular communication; moreover, it may provide a protective role against multiple forms of induced physiological stress [56,60,75,84,86,88,89]. The misfolded, abnormal and insoluble isoform of PrPc known as PrPsc self-associates into pro-inflammatory, protease-resistant aggregates that are insoluble in most detergents and chaotropic agents [59,75,84,89,90,91]. The molecular mechanisms of PrPsc neurotoxicity that drive the initiation, development and progression of PrD are highly complex and, similar to the case of AD, increased oxidative stress and chronic inflammation appear to be critically involved in the initiation and progression of PrD [5,73,86,92,93,94,95]. Typically, activated microglia accumulate within the immediate vicinity of abnormal PrPsc aggregates, and they release cytokines such as IL-1β that play important roles in the inflammatory pathogenesis of PrD, including the upregulation of genes that promote pro-inflammatory signaling and innate-immune system deficits [86,90]. As PrPsc aggregates progressively form over time, they further induce inevitably fatal neurodegenerative disease conditions, including neuroinflammation, which is typically discernable by massive microglial activation and proliferation and the subsequent and self-reinforcing upregulation of cytokines, such as TNF-alpha (TNFα), interleukin 1 alpha (IL-1α) and glial fibrillary acidic protein (GFAP) as well as astrogliosis accompanied by multiple additional pathogenic alterations in the neuronal transcriptome [25,59,67,85,86,96]. The unique primary amino acid sequence of the heavily glycosylated cellular prion PrPc sialoglycoprotein appears to predispose this relatively small molecule to non-homeostatic and pathological secondary and/or tertiary folding changing or ‘flipping’ its spatial conformation into the disease-causing PrPsc protease-resistant isotype [56,75,97]. An analogous situation may occur in the unique primary amino acid sequence of the 42 amino acid human amyloid-beta (Aβ42) peptide that also accumulates as a protease-resistant polymer that progressively accumulates in AD-affected brain [98,99].

Human prion diseases currently include Creutzfeldt–Jacob disease (CJD), stratified into sporadic CJD (sCJD) and variant CJD (vCJD) clinical subtypes, Gerstmann–Sträussler–Scheinker syndrome (GSS), fatal familial insomnia (FFI) and kuru, a fatal neurological disease found among natives from New Guinea who practiced a form of ritual cannibalism in which they consumed the brains of their predecessors [56,57,58,59,60,97]. Animal prion diseases include scrapie in sheep and goats (Ruminants of the order *Artiodactyla*, family *Bovidae*, subfamily *Caprinae*), bovine spongiform encephalopathy (BSE, also known as mad cow disease) in cattle (also of the family *Bovidae*), chronic wasting disease (CWD) in cervids (family *Cervidae*), transmissible mink encephalopathy (TME) in mink (family *Mustelidae*) and feline spongiform encephalopathy (FSE) in cats (family *Felidae;* [56,67,75]). Atypical and novel human prion diseases in *Chordata* continue to emerge, such as the recently identified camel prion disease (CPD) in dromedary camels observed for the first time in Algeria (order *Artiodactyla*, family *Camelidae*) [100]). Prion diseases can also be experimentally studied via the inoculation of brain and other PrPsc-containing extracts into laboratory animals such as mice, voles, gerbils and hamsters, causing a recapitulation of the PrD and TLE in sensitive animals that can be further studied, analyzed and carefully investigated in a biohazard safety level 2 or 3 (BSL-2 or BSL-3) laboratory (see Handling Prions-Environmental Health and Safety, Michigan State University (msu.edu); www.ehs.msu.edu/lab-clinic/bio/handling-prions.html; last accessed on 23 August 2021; [25,67,82,83,84,90]; (Table 1).

Shortly after the first reports of a significantly upregulated miRNA-146a in AD-affected brain and IL-1β-, TNFα- and/or Aβ42 peptide-stressed human neuronal–glial (HNG) cells (transplantation grade) in primary co-culture, the increased abundance of this same pro-inflammatory sncRNA was reported by multiple groups in animal and human nervous tissues affected with PrD [10,54,55,63,75,101]. As a pro-inflammatory sncRNA, over the last ~15 years, miRNA-146a has been repeatedly shown to participate in the regulation of adaptive and innate-immune systems and cytokine-mediated pro-inflammatory responses that potentially culminate in uncontrolled neural tissue damage [24,33,67,69,102,103]. miRNA-146a upregulation in transfected co-cultures of neuronal–glial cells can downregulate both CFH mRNA and protein levels via miRNA-146a pairing with 3′-UTR of human CFH, a finding also observed in multiple murine transgenic models for neural degeneration and in human AD, AMD, MS and TLE [10,22,24,104]. Given that CFH downregulation in neural tissues plays an important role in complement system regulation, immunological signaling and neural cell demise, and since miRNA-146a is the main sncRNA-regulator of neuro-inflammatory responses in neural tissues: (**i**) it is generally accepted that miRNA-146a is critical in the pathogenesis of inflammatory neurodegeneration in multiple forms of immune-related prion diseases, AD and neurotrophic viral infection; and (**ii**) this may be informative for using this microRNA as a general early diagnostic biomarker for multiple forms of these insidious brain diseases [2,4,55].

## 5. Unanswered Questions—Looking Backward and Forward

Since the initial description of scrapie in goats and sheep in about 1750 AD [57,105], there has been a steady emergence of the recognition, identification and characterization of a series of TSEs that now include (in alphabetical order) BSE, CPD, CWD and TME in animals and BSE-like disease, CJD, FFI, GSS and kuru in humans [25,56,60,72,75,82,96,106]. It has been about 64 years since kuru was initially reported to Western medicine, the first described TSE of humans for which the world renown anthropologist, biochemist and virologist D. Carleton Gajdusek was awarded the Nobel Prize in Medicine in 1976 [57,60,72]. At that time, both scrapie and kuru were thought to be caused by a ‘slow virus’ or ‘infectious protein’ that took many years or decades for these lethal neurological disorders to develop in the mammalian brain and CNS [56,59,105]. Twenty-one years later, the biochemist and neurologist Stanley B. Prusiner was awarded the Nobel prize in Medicine in 1997 for the isolation, characterization and proof of transmissibility of novel nucleic-acid-free ‘prion particles’ in susceptible animal models [25,56,72,75]. The relatively recent discovery of camel prion disease (CPD; ‘mad camel disease’) in the Middle East suggests: (**i**) that there may be yet other novel TSEs of animals and humans awaiting our discovery and/or; (**ii**) that mammalian central nervous systems are still in the process of evolving prion-like entities in the expanding spectrum of prion-induced disease [56,75,100]. Serious incapacitating, progressive, age-related and lethal neurological diseases such as AD, first described by the neurologist and physician Alois Alzheimer in 1906, have been suggested to be caused by transmissible, ‘infectious’ prion particles, which, depending on multiple biophysical factors, may adopt alternative conformations that are both self-propagating and found in a very wide array of organisms ranging from yeast to humans [56,75].

Many questions remain unanswered concerning the role of the inducible host miRNA-146a-5p that is upregulated during prion infection as well as by many other types of DNA and RNA viruses and neurodegenerative disease syndromes that include AMD, AD, ALS, CVD, peripheral neuropathies and tumors of the CNS, TLE and TBI (Table 1). Interestingly, the related human microRNA hsa-miRNA-146a-3p does not appear to be significantly upregulated under similar conditions. Our understanding of the potential pathological role of miRNA-146a in AD, prion disease and/or neurotropic viral infection began only about 15 years ago [54,55,61,101]. The list of miRNA-146a participation in prion disease, viral infections of the brain and CNS and related neurodegenerative diseases such as AD is an expanding one. Very fundamental neurobiological questions requiring additional research investigation include the following: (**i**) Does miRNA-146a always contribute to neuro-pathological, neuro-inflammatory and altered neuro-immunological aspects of PrD, viral infection and AD, and/or are other sncRNAs or transmissible particulate species involved?; (**ii**) As pathological and molecular genetic processes associated with neurological disorders often precede clinical symptoms, might miRNA-146a be useful as an early clinical diagnostic and/or prognostic biomarker for viral and/or prion disease and/or AD or other related neurological disorders?; (**iii**) Is the rapid increase in miRNA-146a upon prion or viral infection advantageous to the host via an innate-immunity-mediated mechanism, are upregulated miRNAs part of the invading prion or a viral strategy for a more efficient invasion and infection, or are both of these biomolecular and/or immunological scenarios plausible?; (**iv**) Would both of these possibilities be expected to involve miRNA-146a-specific mRNA targets and modulation of gene expression signaling to ultimately alter the transcriptome of the neuron and/or neural support cells?; (**v**) Are other transcription factors besides NF-κB and other pro-inflammatory microRNAs besides miRNA-146a involved in initiating, driving or modulating prion- and/or viral-directed neuro-degeneration?; (**vi**) Are anti-NF-kB (p50/p65)- and/or anti-miRNA-146a-5p-based therapeutic strategies clinically feasible to be deployed, and are they suitable to address and successfully treat the broad spectrum of human neurological disorders and other neurodegenerative disease syndromes initiated by neurotropic DNA and RNA viruses and/or prions?; and (**vii**) Would these same pharmacological strategies and treatments be useful in the clinical management of viral infection, prion disease, AD and/or perhaps other progressive, neurological disorders associated with aging and progressive degeneration of the brain and CNS?

## 6. Conclusions 

Among the most significant advances in human neuroscience, neurology and molecular neurogenetics over the last fifteen years are: (**i**) the discovery of a family of small noncoding single-stranded RNAs called microRNAs in the mammalian brain and central nervous system (CNS) and (**ii**) the analysis and categorization of their abundance, speciation and complexity in development, aging and in neurological health and CNS disease [2,5,6,9,70,71,93,101,107]. A growing body of evidence indicates that select species of the 2650 member human miRNA gene family are brain-abundant and participate in the initiation, propagation and development of insidious age-related neurological disorders of the mammalian brain and CNS. This includes the involvement of a unique pro-inflammatory miRNA-146a in a broad spectrum of viral- and prion-induced encephalopathies and related progressive age-related neurodegenerations of the human brain that include, prominently, AD, ALS, AMD, MS, TLE, scrapie and BSE (mad cow disease) as well as CJD, GSS and kuru. miRNA-146a’s role and significance in viral-induced encephalopathies and prion disease appear to be expanding. Several attractive and all-encompassing recently proposed theories suggest: (**i**) that there is a contribution of prions and misfolded proteins in human degenerative diseases that include AD and AMD and other miRNA-146a-associated neurological disorders [56,75,108]; and (**ii**) that there may be a gastrointestinal (GI)-tract-sourced contribution of microbes or microbial neurotoxins to AD and related neurodegenerative disorders that critically involve miRNA-146a-mediated immunological and/or pro-inflammatory signaling components [108,109,110,111].

## Figures and Tables

**Table 1 ijms-22-09198-t001:** Progressive, age-related neurological disorders in which miRNA-146a-5p is significantly upregulated in brain or CNS tissues; all neurotropic viruses indicated ultimately affect human brain or CNS function; (−)ssRNA = negative single-stranded RNA; (+)ssRNA = positive single-stranded RNA; dsDNA = double-stranded DNA genomes; prions contain no detectable nucleic acids [25]; natural hosts for prion infection are indicated; ‘interspecies transmission’ indicates disease transmissibility in experimental laboratory animals; there may be other neurotropic viruses or PrDs that involve miRNA-146a-5p remaining to be discovered; all neurotropic virus and prion diseases are alphabetically ordered; see also manuscript text.

**Neurotropic Viral Pathogen**	**Genus**	**Genome**	**Reference**
Borna encephalitis disease virus 1 (BoEDV-1; BDV)	*Mononegavirales*	(−)ssRNA	[26,27]
Chikungunya virus (CHIKV)	*Togaviridae*	(+)ssRNA	[28]
enterovirus 71 (EV71)	*Picornaviridae*	(+)ssRNA	[29]
Epstein-Barr virus (EBV)	*Herpesviridae*	dsDNA	[30]
Hantavirus (HTV)	*Bunyaviridae*	(−)ssRNA	[31]
hepatitis A virus (HAV)	*Picornaviridae*	(+)ssRNA	[32]
hepatitis B virus (HBV)	*Hepadnaviridae*	dsDNA	[33,34,35,36]
hepatitis C virus (HCV)	*Flaviviridae*	(+)ssRNA	[37];
herpes simplex virus-1 (HSV-1)	*Herpesviridae*	dsDNA	[21,38,39,40,41]
Hendra virus (Henipavirus; HeV)	*Paramyxoviridae*	(−)ssRNA	[42]
human immunodeficiency virus (HIV)	*Retrovirida*	(+)ssRNA	[43]
human influenza A viruses (H1N1/H3N2)	*Orthomyxoviridae*	(+)ssRNA	[44,45]
early human papillomavirus virus 16 (eHPV-16)	*Papillomaviridae*	dsDNA	[46]
SARS-CoV-2 (*agent for COVID-19 disease)*	*Betacoronavirus*	*(+)ssRNA*	[47]
human T-cell leukemia virus type 1 (HTLV-1)	*Retroviridae*	(+)ssRNA	[48]
Japanese encephalitis virus (JEV)	*Flaviviridae*	(+)ssRNA	[49,50]
Kaposi’s sarcoma-associated herpesvirus (KSHV)	*Herpesviridae*	dsDNA	[51]
severe fever-thrombocytopenia syndrome virus (SFTSV)	*Bunyaviridae*	(−)ssRNA	[52]
**neurological disease** **(natural or experimental)**	**natural host (family)**	**interspecies transmission**	**Reference**
**prion disease (PrD)**			
bovine spongiform encephalopathy (BSE)	*Bovidae; Hominidae*	+	[16,41,53,54]
chronic wasting disease (CWD)	*Cervidae*	?	[55]; Pogue & Lukiw, unpublished
sporadic Creutzfeldt–Jacob disease (sCJD)	*Hominidae*	+	[10,13,14,56,57]
variant Creutzfeldt–Jacob disease (sCJD)	*Hominidae*	+	[10,13,14,56,57,58]
Gerstmann–Sträussler–Scheinker syndrome (GSS)	*Hominidae*	+	[10,13,14,56]
fatal familial insomnia (FFI)	*Hominidae*	+	[16,53]; Pogue & Lukiw, unpublished
Kuru	*Hominidae*	+	[59,60]
murine scrapie (experimental)	*Muridae*	+	[14,54,61]
transmissible mink encephalopathy (TME)	*Mustelidae*	+	[56]; Pogue & Lukiw, unpublished
feline spongiform encephalopathy (FSE)	*Felidae*	?	Pogue and Lukiw, unpublished
camel prion disease (CPD)	*Camelidae*	?	[2,56]
**human neurodegenerative disease**			
age-related macular degeneration (AMD)	*Hominidae*	?	[11,21,39]
Alzheimer’s disease (AD)	*Hominidae*	?	[3,10,21,40,62,63]
amyotrophic lateral sclerosis (ALS)	*Hominidae*	?	[3,23]
cerebrovascular disease (CVD)	*Hominidae*	?	[3,18]
peripheral neuropathies/tumors of the CNS	*Hominidae*	?	[3,24]
temporal lobe epilepsy (TLE)	*Hominidae*	?	[20]
traumatic brain injury (TBI)	*Hominidae*	?	[19]

## Data Availability

All data used in this review are openly available and freely accessible on MedLine (www.ncbi.nlm.nih.gov accessed on 16 August 2021) where they are listed by the last names of the individual authors.

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
