# Peer review of "microRNA-146a-5p, Neurotropic Viral Infection and Prion Disease (PrD)"

_ijms, 2021, doi:10.3390/ijms22179198_

Round 1

Reviewer 1 Report

The authors mainly summarized the Neurotropic viral-mediated induction of hsa-miRNA-146a-5p, the upregulated effect of Prion disease on hsa-miRNA-146a-5p, as well as some future research directions. This manuscript is helpful for an in-depth study of “microRNA-146a-5p, neurotropic viral infection and prion disease”.

Some comments:

  1. Line 59, the authors stated “the possibility of 4 ribonucleotides at each position (A,C,T or U; adenine, cytosine, thymine or uracil)”. No guanine (G)? Here the author needs to add citations.
  2. miRNA-146a is a previous name and it is recommended to change it to “hsa-miRNA-146a-5p”. And why did the authors not mention the closely related hsa-miRNA-146a-3p.
  3. Line 80, the authors stated “Homo sapien microRNA-146a-5p (hsa-miRNA-146a; miRNA-146a) whose  significant  up-regulation  is  currently  implicated  in  every  prion disease  (PrD)  in  humans  and  animals”. Did hsa-miRNA-146a-5p exist in animals?? The same confusion is in line 314.
  4. Line 95, MI0000477 is not accurate. MIMAT0000449 is right.
  5. Line 105, The authors need to provide the version of the human genome. GRCh38 (hg38)?
  6. It is strongly recommended that the authors add corresponding diagrams and/or tables in each section to make it easier for the readers to read.

Author Response

REVIEWER 1

Reviewer 1 makes the following comments:

English language and style

( ) Extensive editing of English language and style required
( ) Moderate English changes required
(x) English language and style are fine/minor spell check required
( ) I don't feel qualified to judge about the English language and style

[NOTE: a maximum of 5 stars signifies excellence and the highest score available

Is the work a significant contribution to the field? ****

Is the work well organized and comprehensively described? ***

Is the work scientifically sound and not misleading? ****

Are there appropriate and adequate references to related and previous work? *****

Is the English used correct and readable? *****          

Firstly we would like to thank Reviewer 1 for their positive and supportive comments: ‘that the work (was) a significant contribution to the field’; that ‘the work (was) scientifically sound and not misleading; that ‘there (were) appropriate and adequate references to related and previous work’; and that ‘the English used (was) correct and readable’;

The authors mainly summarized the Neurotropic viral-mediated induction of hsa-miRNA-146a-5p, the upregulated effect of Prion disease on hsa-miRNA-146a-5p, as well as some future research directions. This manuscript is helpful for an in-depth study of “microRNA-146a-5p, neurotropic viral infection and prion disease”.

Some comments:

COMMENT:

  1. Line 59, the authors stated “the possibility of 4 ribonucleotides at each position (A,C,T or U; adenine, cytosine, thymine or uracil)”. No guanine (G)? Here the author needs to add citations.

RESPONSE: Thank you for pointing out this oversight on our part. This has been corrected; two web links have been added in the revised manuscript text; as stated in the manuscript text this is a straightforward mathematical operation/calculation.

=====================

 COMMENT:

  1. miRNA-146a is a previous name and it is recommended to change it to “hsa-miRNA-146a-5p”. And why did the authors not mention the closely related hsa-miRNA-146a-3p.

RESPONSE:

As indicated on lines 82-83 the manuscript text the abbreviated form of Homo sapien microRNA-146a-5p (hsa-miRNA-146a-5p), as is common in the scientific literature, is just written as ‘miRNA-146a’. If there is a deviation from this anywhere in the manuscript text it is so stated. A note on lines 359-361 in the revised manuscript text mention that the closely related hsa-miRNA-146a-3p is not significantly up-regulated in the viral, prion or related disorders discussed in this review paper,. As should be clear the major focus of this review paper is the up-regulation of hsa-miRNA-146a-5p in the multiple neurological disorders.

=====================

COMMENT:

  1. Line 80, the authors stated “Homo sapien microRNA-146a-5p (hsa-miRNA-146a; miRNA-146a) whose  significant  up-regulation  is  currently  implicated  in  every  prion disease  (PrD)  in  humans  and  animals”. Did hsa-miRNA-146a-5p exist in animals?? The same confusion is in line 314.

RESPONSE:

As already stated in the manuscript text (lines 94-99), the ribonucleotide sequence of hsa-miRNA-146a-5p is identical in human and mouse; evolutionary and experimental-research aspects and implications are briefly discussed; while miRNA-146a has been detected in several other animals (via the use of antibodies and fluorescent array panels) its RNA sequence in most animal species has yet to be determined;

=====================

COMMENT:

  1. Line 95, MI0000477 is not accurate. MIMAT0000449 is right.

RESPONSE:

Thank you for pointing this out. This term and web link have been corrected in the revised manuscript text

=====================

COMMENT:

  1. Line 105, The authors need to provide the version of the human genome. GRCh38 (hg38)?

RESPONSE:

It is GRCh38/hg38; this version has been provided in the web link already provided (https://www.genecards.org/cgi-bin/carddisp.pl?gene=MIR146A) and is now stated in the revised manuscript text.

=====================

COMMENT:

  1. It is strongly recommended that the authors add corresponding diagrams and/or tables in each section to make it easier for the readers to read.

RESPONSE: Via the use of multiple specific and recent hsa-miRNA-146a-5p references we chose to refer to the many Figures that have already been peer-reviewed and published regarding multiple aspects of hsa-miRNA-146a-5p genetics and neurobiology to avoid reiteration of the same data; 

However to allay Reviewer 1’s concerns we have now included a comprehensive Table 1 which indicates miRNA-146a-5p’s involvement in various prion- and viral mediated disorders and related progressive neurodegenerative diseases.

=====================

RESPONSE TO REVIEW

We would like to kindly and sincerely thank Reviewer 1 for their valuable time and expertise in the examination of our current review paper.

Their insightful and constructive comments and addition of Table 1 and one highly relevant new reference have resulted in a clarified and strengthened contribution for the International Journal of Molecular Sciences (IJMS).

Yours truly

Walter J. Lukiw BS, MS, PhD, Professor of Neurology, Neuroscience and Ophthalmology, Bollinger Professor of Alzheimer’s disease (AD), LSU Neuroscience Center, Louisiana State University Health Sciences Center, 2020 Gravier Street, Suite 904, New Orleans LA 70112 USA, TEL 504-599-0842, EMAIL [email protected]

Reviewer 2 Report

This subject has been the focus of a more thorough recent review here: Fan, W.; Liang, C.; Ou, M.; Zou, T.; Sun, F.; Zhou, H.; Cui, L. Microrna-146a is a wide-reaching neuroinflammatory regulator and potential treatment target in neurological diseases. Front Mol Neurosci 2020, 13, 90-90.

The article presented does not represent a substantive increase in the current knowledge on this subject.

Author Response

REVIEWER 2

Reviewer 2 makes the following comments:

English language and style

( ) Extensive editing of English language and style required
( ) Moderate English changes required
(x) English language and style are fine/minor spell check required
( ) I don't feel qualified to judge about the English language and style

[NOTE: a maximum of 5 stars signifies excellence and the highest score available

Is the work a significant contribution to the field? *

Is the work well organized and comprehensively described? ****

Is the work scientifically sound and not misleading? ****

Are there appropriate and adequate references to related and previous work? *

Is the English used correct and readable? *****

COMMENT:

This subject has been the focus of a more thorough recent review here: Fan, W.; Liang, .; Ou, M.; Zou, T.; Sun, F.; Zhou, H.; Cui, L. Microrna-146a is a wide-reaching neuro-inflammatory regulator and potential treatment target in neurological diseases. Front Mol Neurosci 2020, 13, 90-90. The article presented does not represent a substantive increase in the current knowledge on this subject.

RESPONSE:

We thank Reviewer 2 for their positive comments that ‘the work (was) well organized and comprehensively described’; that ‘the work (was) scientifically sound and not misleading’; and that ‘the English used (was) correct and readable’;

Regarding Reviewer #2’s comment on the Fan et al, 2020 paper:

(i) We are well familiar with the (above mentioned) Fan et al., 2020 paper quoted above. In fact, this Fan et al., 2020 paper has already been quoted in the Reference list of our current manuscript. However, this is a very rapidly evolving area of neurodegenerative disease research and microRNA research and many of the reference papers in the Fan et al., 2020 article are well out of date. From information available on Medline and other available databases, data for the Fan et al., 2020 review paper was complied in late 2019; was received by the journal (Front Mol Neurosci) February 2020 – this is greater than 18 months or a year-and-a-half ago;

(ii) Put another way our paper is a considerably more comprehensive and up-to-date review of the role of miRNA-146a in viral, prion and related neurodegenerative diseases such as Alzheimer’s disease (AD). Just to quote one example, the Fan et al., 2020 review contains just one 2020 paper and zero 2021 references while our current review paper contains twelve 2020 reference papers and twenty 2021 references;

(iii) Importantly, what the Fan et al., 2020 paper doesn’t mention are the huge growing and recent body of research work and evidence which describes the up-regulation of miRNA-146a and its extensive involvement in at least 18 different neurotropic viral diseases including, importantly, SARS-CoV-2, the causative agent of COVID-19 (a finding in late 2020-early 2021); much of the data in the Fan et al., 2020 is incomplete; they mention only three viral infections in which miRNA-146a is involved; JEV, HSV, HIV; we review the more extensive evidence of eighteen viral infections that appear to involve an up-regulation of miRNA-146a at the core of the neurodegenerative disease mechanism;

(iv) in addition, what is not mentioned anywhere in the Fan et al 2020 paper is that miRNA-146a up-regulation is shared by multiple human neurological disease mechanisms and there may be some underlying commonality to miRNA-146a’s role in inflammatory neurodegeneration across a broad spectrum of terminal human brain and CNS disease and in experimental animal models;

(v) the authors of our current review paper were the very first to publish on the involvement of miRNA-146a in neurodegenerative brain disease (Alzheimer’s disease; Lukiw WJ, Zhao Y, Cui JG. An NF-kB-sensitive micro RNA-146a-mediated inflammatory circuit in Alzheimer disease and in stressed human brain cells. J Biol Chem. 2008) and has been the major research focus of our laboratory for over the last  ~15 years. The first (post-doc) author of this paper AI Pogue has carefully collected and comprehensively organized all relevant miRNA-146a data over the last decade-and-a-half with emphasis on the last several years, integrating all of the most recent data into the current review paper.

(vi) we further note that a large part of the current data on miRNA-146a-5p in neurological disease is presented in a new comprehensive full page Table 1 in our current revised review paper that is not available anywhere else in the current literature.

Reviewer 3 Report

It is an excellent review of current knowledge about miR-146a-5b and its potential involvement in neurodegenerative diseases.

Author Response

Reviewer 3 makes the following comments:

English language and style

( ) Extensive editing of English language and style required
( ) Moderate English changes required
( ) English language and style are fine/minor spell check required
(x) I don't feel qualified to judge about the English language and style

[NOTE: a maximum of 5 stars signifies excellence and the highest score available]

Is the work a significant contribution to the field? *****

Is the work well organized and comprehensively described? *****

Is the work scientifically sound and not misleading? *****

Are there appropriate and adequate references to related and previous work? *****

Is the English used correct and readable? *****          

COMMENT:

Comments and Suggestions for Authors

It is an excellent review of current knowledge about miR-146a-5b and its potential involvement in neurodegenerative diseases.

RESPONSE:

We sincerely thank Reviewer #3 for their highly positive comment that this paper ‘is an excellent review of current knowledge about miR-146a-5b and its potential involvement in neurodegenerative diseases.’

=====================

RESPONSE TO REVIEW

We would again like to sincerely thank Reviewer 3 for their valuable time and expertise in the examination of our current review paper and their comments on our review that this paper was ‘an excellent review of current knowledge about miR-146a-5b and its potential involvement in neurodegenerative diseases.’

All authors sincerely believe that this manuscript represents a current, in depth and up-to-date review of the role of miRNA-146a-5p in neurobiology and neuropathology and would be an interesting and well-received contribution to readers of the International Journal of Molecular Sciences (IJMS). The addition of Table 1 has further resulted in a strengthened contribution for the International Journal of Molecular Sciences (IJMS).

Thank you kindly,

Yours truly

Walter J. Lukiw BS, MS, PhD, Professor of Neurology, Neuroscience and Ophthalmology, Bollinger Professor of Alzheimer’s disease (AD), LSU Neuroscience Center, Louisiana State University Health Sciences Center, 2020 Gravier Street, Suite 904, New Orleans LA 70112 USA, TEL 504-599-0842, EMAIL [email protected]

=======================================================================

Round 2

Reviewer 1 Report

no comments

Reviewer 2 Report

Thank you to the authors for clarifying comments in their response and subsequent revision to this document.